# The Efficacy of Tumor Mutation Burden as a Biomarker of Response to Immune Checkpoint Inhibitors

**DOI:** 10.3390/ijms24076710

**Published:** 2023-04-04

**Authors:** Camille Moeckel, Katrina Bakhl, Ilias Georgakopoulos-Soares, Apostolos Zaravinos

**Affiliations:** 1Department of Biochemistry and Molecular Biology, Institute for Personalized Medicine, The Pennsylvania State University College of Medicine, Hershey, PA 17033, USA; 2Department of Life Sciences, European University Cyprus, Diogenis Str., 6, Nicosia 2404, Cyprus; 3Cancer Genetics, Genomics and Systems Biology Laboratory, Basic and Translational Cancer Research Center (BTCRC), Nicosia 1516, Cyprus

**Keywords:** tumor mutation burden, tumor microenvironment, immune checkpoint inhibitors, immunotherapy

## Abstract

Cancer is one of the leading causes of death in the world; therefore, extensive research has been dedicated to exploring potential therapeutics, including immune checkpoint inhibitors (ICIs). Initially, programmed-death ligand-1 was the biomarker utilized to predict the efficacy of ICIs. However, its heterogeneous expression in the tumor microenvironment, which is critical to cancer progression, promoted the exploration of the tumor mutation burden (TMB). Research in various cancers, such as melanoma and lung cancer, has shown an association between high TMB and response to ICIs, increasing its predictive value. However, the TMB has failed to predict ICI response in numerous other cancers. Therefore, future research is needed to analyze the variations between cancer types and establish TMB cutoffs in order to create a more standardized methodology for using the TMB clinically. In this review, we aim to explore current research on the efficacy of the TMB as a biomarker, discuss current approaches to overcoming immunoresistance to ICIs, and highlight new trends in the field such as liquid biopsies, next generation sequencing, chimeric antigen receptor T-cell therapy, and personalized tumor vaccines.

## 1. Introduction

Currently, cancer is one of the top three causes of death in the world [1], but immune checkpoint inhibitors (ICIs) have demonstrated potential in treating a multitude of these malignancies, including melanoma and non-small cell lung cancer (NSCLC) [2]. Nevertheless, the discovery of predictive biomarkers of the efficacy of these ICIs has been complicated. Therefore, it has been crucial to explore new biomarkers including the tumor mutation burden (TMB).

Initially, the TMB was defined by whole exome sequencing of both the tumor’s DNA and the corresponding normal DNA in order to exclude usual germline alterations in the DNA sequence [3]. TMB was subsequently classified as the overall number of somatic or coding mutations; in specific cases, the estimate included deletions and insertions. Although whole genome sequencing is the optimal measurement method for TMB, this methodology is not routinely used clinically due to both cost and complexity [2]. Instead, TMB is clinically defined as the total sum of base substitutions in targeted genes’ coding regions; for instance, targeted tumor-sequencing panels, such as the MSK-IMPACT^®^ (Memorial Sloan Kettering-Integrated Mutation Profiling of Actionable Cancer Targets) can be utilized [4]. MSK-IMPACT^®^ is the first laboratory-developed tumor-profiling test to receive authorization from the U.S. Food and Drug Administration. It can detect greater than 468 gene mutations and other critical genetic changes, such as microsatellite instability, in common and rare cancers, to help with the identification of those patients who may benefit from immunotherapy. However, there is no standard for TMB measurement, and logistical issues can get in the way of implementing TMB evaluation in daily clinical practice [5].

Before TMB was investigated as a biomarker for ICI efficacy, research focused on programmed-death ligand-1 (PDL-1) expression in the tumor microenvironment (TME); the binding of programmed death cell receptor-1 (PD-1) to PD-L1 limits activation of T cells and subsequently decreases the immune response to cancer cells [6]. Studies have demonstrated the success of PD-L1 expression in determining specific patients expected to benefit from ICIs, and quantitation was endorsed as a diagnostic for treatment of NSCLC patients with pembrolizumab [7]. However, due partly to PD-L1’s heterogeneous expression in the TME, this biomarker’s relationship with ICI response is inconsistent [6].

As a result of inconsistencies with other biomarkers such as the aforementioned PD-L1 expression, research has strived to evaluate the potential for TMB as a biomarker [8]. It was initially hypothesized to act as a biomarker because when the number of somatic mutations is increased, more neoantigens are generated, and there is a higher chance that one or more will be immunogenic and subsequently prompt a T cell reaction [9]. Recently, a multitude of studies have demonstrated that in cancer patients treated with ICIs, a high TMB is significantly correlated with improved prognosis [10,11]. However, the TMB has failed to predict response to immunotherapies in other cancer diagnoses, such as breast and prostate cancers and glioma [12]. Therefore, it is crucial that research continues to investigate the importance of TMB to immunotherapeutic treatment for numerous cancers.

## 2. Cell Composition in the Tumor Microenvironment

Within an individual’s tissues, tumor cells continuously induce important molecular and physical changes, and the evolving TME is critical to cancer progression. Although the TME’s composition varies by cancer type and patient [13], it consistently develops surrounding tumor cells and is composed of the extracellular matrix, immune cells, mesenchymal stromal cells, blood vessels, and fibroblasts (Figure 1A) [14]. The tumor cells can manipulate the surrounding environment to downregulate anti-tumor activity and increase angiogenesis to the site (Figure 1B) [13]. Due to the influence on malignancy growth and metastasis, research on cancer therapeutics has more recently started to focus on targeting the TME.

Stromal cells, which include vascular endothelial cells, adipocytes, fibroblasts, and stellate cells, are a crucial part of the TME [14]. These cells are recruited by cancer cells from nearby endogenous tissue stroma in order to both provide nutrition to the tumor cells and clear away waste and debris from the surroundings. As the tumor increases in size, the TME can become both hypoxic and acidic due to insufficient oxygen and an increase in waste products; however, the hypoxia requires tumors to develop a blood supply, which leads to angiogenesis and further tumorigenesis [15]. Collagen interacts with other components of the extracellular matrix, such as hyaluronic acid, laminin, and fibronectin, in order to both act as a physical backbone for cells and promote cancer cell growth and dissemination [15].

In order to promote tumor progression, fibroblasts in the TME, which are referred to as Cancer-Associated Fibroblasts (CAFs), increase the expression of VEGF to further promote angiogenesis, modulate immune responsiveness, and influence cell metabolism (Figure 1B) [16]. CAFs in the TME produce the vast majority of extracellular components, and their release of hepatocyte growth factor 5, growth differentiation factor 15, fibroblast growth factor 5, and TGF-beta aids in the abnormal proliferation of cancer cells [16]. Specifically, the release of hepatic growth factor 5 has assisted in tumor resistance to therapies targeting the BRAF protein directly by allowing for other pathways that also achieve ERK-MAPK activation. In addition, tumor necrosis factor (TNF) produced by CAFs is able to assist with fibroblast activation [16].

CAFs encourage an immunosuppressive phenotype by producing immune modulatory chemokines and cytokines, which influence the actions of regulatory T cells, CD8+ cells, and macrophages [17]. Specifically, CAFs have the ability to decrease CD8+ T cell response through the secretion of interleukin 6, TGF-beta, and CXC-chemokine [18]. In terms of the CAFs’ influence on metabolism, the “Reverse Warburg effect” has been proposed as a mechanism behind supplying cancer cells with energy for growth and proliferation [19]. According to this model, tumor cells secrete hydrogen peroxide, which induces oxidative stress in the stromal cells; subsequently, CAFs undergo aerobic glycolysis in order to supply nearby cancer cells with high-energy products such as lactic acid, fatty acids, ketone bodies, and pyruvate [19].

In addition to CAFs, immune cells including T cells, B cells, natural killer cells, and macrophages, are crucial components of the TME (Figure 1C). Interestingly, in the TME, immune cells can either have a role in tumor growth or suppression [19]. However, largely, the suppressive functions of immune effector cells recruited to the TME are downregulated by tumor-derived signals [20]. Tumor-infiltrating lymphocytes (TILs) are one of the more prevalent immune components of the anti-tumor response; however, research has shown that when TILs are stimulated by tumor antigens, they show slowed proliferation, weakened signaling through T cell receptors, and decreased capability to produce Th1-type cytokines or mediate cytotoxicity of tumor targets [20].

Myeloid-derived suppressor cells (MDSCs) within the TME also aid in tumor development and immune tolerance. MDSCs typically relocate to the peripheral lymphoid organs and differentiate into macrophages and dendritic cells [21]. However, in the TME, MDSCs differentiate into tumor-associated macrophages (TAMs). TAMs are categorized as M1 or M2; M1 macrophages exhibit anti-tumor responses, while M2 macrophages promote tumor growth through several mechanisms including neovascularization, angiogenesis, and modification of stromal cells for greater tumor support [22,23]. M0 macrophages are undifferentiated macrophages, but are prone to differentiate into M2 macrophages because they are recruited to the site via M2-associated cytokines [22].

Overall, MDSCs boost immunosuppression by blocking the migration of CD 8+ T cells to the tumor, downregulating T cell receptors, decreasing reactive oxygen species (ROS), increasing the hypoxia-inducible factor, and upregulating PD-L1 [21]. In addition, MDSCs increase the production of CCL4 and CCL5 chemokines within the TME, which leads to the increased migration of TAMs and regulatory T cells to the tumor; ultimately, the result is immunosuppressive activity in the TME [24]. Specifically, the regulatory T cells suppress the immune system’s response to tumor cells, while the TAMs drive inflammation through cytokine release, notably of interleukin 23 and interleukin 17, and play a role in tumor metastasis by increasing the likelihood of tumor migration [25].

Recently, research has focused on the impact of both the microbiome and mycobiome on the TME (Figure 1D). Studies have demonstrated that the gut microbiome has an integral role in the anti-tumor immune response and subsequently impacts a patient’s response to ICIs [26,27,28,29]. The gut microbiome’s impact on the TME includes decreasing tumor quantity, decreasing inflammation, and slowing metastatic growth [30]; studies have implicated specific gut microbial metabolites such as TMAO in driving anti-tumor immunity [31]. However, recent studies have also implicated specific microbiota in driving inflammation and particular forms of cancer growth [32,33]. Therefore, it is imperative for research to continue exploring the connections between the microbiome, tumorigenesis, and the response to cancer immunotherapies [27]. This research will promote future strategies to modulate the microbiome to improve outcomes to ICIs [27].

In terms of the mycobiome, a study on pancreatic ductal adenocarcinoma (PDA) has shed light recently on the impact of disbalance in the mycobiome [34]. When compared with healthy pancreatic tissue, a sample of PDA contained a 3000-fold amount of fungi, especially *Malassezia* spp. [34]. However, when ablation of the mycobiome was performed, there was evident protection against tumor proliferation in slowly progressing and invasive forms of PDA [34]. Lastly, the study showed that propagation with a Malassezia species specifically accelerated oncogenesis [34]. In addition, a study analyzing the presence of Candida, a fungus, in gastrointestinal cancers found that several Candida species were increased in tumor samples and tumor-associated Candida DNA was predictive of reduced patient survival [35]. The fungus was implicated in an increased pro-inflammatory immune response and linked with decreased regulation of genes involved in cellular focal adhesion and metastasis [35].

Given the evolving research on the mycobiome’s role in tumor development, it has also been recently explored as a target in cancer therapy. Administration of specific β-glucans, one of the polysaccharides in the fungal cell wall, has been shown to regulate the TME, resulting in a decrease in tumor growth and metastasis [36]. Other research has suggested that β-glucan may modulate the immune response, subsequently increasing the response to ICIs [36]. Studies such as these push forward knowledge about the TME and therapeutic treatment options for cancer patients [36].

## 3. Variability in Mutations across Different Tumors

Despite immunosuppressive TMEs, immunotherapies have recently shown remarkable success in a number of cancer types and in tumors that have inactivation of specific genes [37]. The TMB levels, specifically, vary across cancer patients, with the total number of mutations observed per genome often differing by several orders of magnitude [38,39,40]. Higher mutational load is more representative of malignancies developed due to powerful carcinogens, including tobacco smoke in NSCLCs, and mutagen exposure, including from ultraviolet light in the case of melanoma [38,40]. Alternatively, renal, ovarian, and breast cancers have intermediate TMB levels, while pediatric tumors and leukemias have lower TMB levels. The TMB can also vary widely in one individual cancer type. A patient’s age has been shown to impact the TMB [38], and there is also intratumoral heterogeneity across tumor types [41,42]. Later in a tumor’s evolution, mutations can occur only in select cells; these subclonal mutations have been attributed to the APOBEC cytidine deaminase family [41].

Germline inactivation of specific genes can additionally result in a higher TMB. Microsatellite instability, which is caused by impaired DNA mismatch repair in genes including MLH1, MSH2, MSH6, and PMS2, is associated with high TMB across cancer types [43]. Deficient mismatch repair can lead to mutations in the DNA polymerase genes [44], and both germline and somatic mutations in key polymerases, such as POLD1 and POLE, have been shown to also result in a higher TMB [45]. In addition, homologous recombination deficiency has been linked to a higher neoantigen load [46].

Although recent evidence has suggested that higher somatic TMB (highest 20% in each histology) is significantly associated with better outcomes across cancer types following ICIs, the TMB cut points vary distinctly between disparate cancer types [47]. Furthermore, research has demonstrated that the levels of various types of mutations can differently influence the success of ICIs. For instance, tumor aneuploidy is associated with immune evasion and a diminished response to immunotherapy [48]. On the other hand, insertions and deletions can result in neoepitopes and increased immunogenicity [49]; this holds true especially for frameshift mutations, which are highly immunogenic [50].

Mutations in specific genes and pathways have also been associated with patients’ responses to immunotherapies. For example, mutations in anaplastic lymphoma kinase (ALK) and epidermal growth factor receptor (EGFR) result in a reduced response rate to ICIs [51], whereas the loss of ADAR1 overcomes resistance to the PD-1 checkpoint blockade [52]. In addition, the inactivation of the interferon-γ pathway genes is associated with a worse response to ICIs [53], while mutations in the NOTCH signaling pathway can result in improved response [54]. Lastly, aberrant expression of endogenous retroviruses has also been associated with improved response to immunotherapy [55]. It is ultimately important to thoroughly understand how both germline-derived and somatic mutations change the cellular phenotype and allow tumor cells to proliferate and spread despite normal physiological constraints.

## 4. The Association between Mutations and the Immune System Response

As previously stated, a critical aspect of both the development and pathogenesis of cancer is the immune system’s inability to identify and subsequently eradicate cancer cells [56]. A small subset of somatic mutations in the cancer cells’ DNA can, however, produce mutation-derived neoantigens, which can be targeted by the immune system (Figure 2A,B) [57,58,59]. Although all mutations will not result in neoantigen production, when a tumor has more somatic mutations, more neoantigens are formed [9,60].

After transcription and translation of the genes containing the mutations, the neoantigen-containing peptides undergo processing and are subsequently presented on major histocompatibility complex (MHC) molecules on the cell surface. Neoantigens, unlike autoantigens, are not predisposed to immune tolerance; instead, they are identified by the autoimmune system, resulting in T cell activation and an immune response [57,61]. The more different a neoantigen is to the healthy proteome, the more immunogenic it tends to be [62].

Tumor cells can evade immune recognition and reduce the cytotoxic T cell response against transformed cells despite neoepitope presentation due in part to the TME’s immunosuppressive features. Cancer cells are able to mobilize regulatory T cells (Tregs), decrease expression of tumor antigens, and ultimately, bring about T cell tolerance and/or apoptosis (Figure 2C) [63]. Recent research has shown that an abundance of interferon-gamma, specifically in the lung lymph nodes, induces suppressive Th1-like effector Treg cells, which subsequently interact with dendritic cells to prevent cytotoxic T cell responses against lung cancer [64]. Additionally, research has indicated that tumor cells have the ability to release immunosuppressive cytokines, which ultimately stimulate inhibitory immune checkpoints and create an immunosuppressive TME [20,63].

Immune checkpoints are defined as stimulatory or inhibitory pathways that preserve self-tolerance and help with the immune system’s response (Figure 2D) [65]. In homeostatic conditions, immune checkpoints maintain the balance between proinflammatory and anti-inflammatory signaling; tumor cells, however, disrupt the homeostasis to promote an immunosuppressive state which allows for evasion of the immune system and further tumor growth [65]. The most well-described inhibitory checkpoints include CTLA-4 or cytotoxic T lymphocyte-associated molecule-4, PD-1 or programmed cell death receptor-1, and PD-L1 or programmed cell death ligand-1. CTLA-4 is upregulated on active T cells’ surfaces in order to inhibit excessive activation by T cell receptors, which activate T cells [66]. PD-1, which is additionally upregulated on activated T cells, binds to PD-L1 and subsequently transmits a negative costimulatory signal to limit activation of T cells [66]. The TME is classically distinguished by PD-L1 overexpression by tumor cells and CTLA-4 and PD-1 overexpression by T cells [67]. Ultimately, this phenotype enables inhibitory checkpoint signals to limit T cell activation and allows tumors to avoid immune surveillance.

## 5. Current Immune Checkpoint Inhibitors and Beyond

In the recent past, the three crucial components of cancer treatment were surgery, chemotherapy, and radiation; all three aim to reduce tumor-burden-associated impacts on immunity by reducing the tumor size and adjusting the TME to hopefully alleviate immune suppression [68]. In 2011, however, ipilimumab, which is an anti-CTLA-4 monoclonal antibody, was FDA-approved as the first ICI for treatment of advanced melanoma [69]. Since then, ICIs, which block a selected inhibitory pathway’s effects in order to overcome immunosuppressive conditions, have become the standard of care when treating many malignancies (Figure 2E) [63,65,70]. Additional ICIs were developed, including anti-PD-1 agents, such as pembrolizumab, nivolumab, and cemiplimab, and anti-PD-L1 agents, such as avelumab, atezolizumab, and durvalumab [66].

ICIs strive to intensify the host immune system’s response to tumor cells and can subsequently lead to tumor remission in a small subsection of patients [2]. For example, in the case of NSCLC, studies have demonstrated the benefit of ICIs in patients who do not have active mutations in ROS-1 or EGFR [71]. However, in most cancers, the response rate to ICIs is only 15–40% [72]. Studies have hypothesized that the patients who do not benefit from this treatment type have resistance due to inadequate or lacking anti-tumor immune responses.

Since the majority of cancer patients do not have tumor remission following ICIs, researchers have attempted combination strategies; for example, anti-CTLA-4 agents have been utilized along with anti-PD-1 and PD-L1 [66,67]. However, with combination strategies, there is concern for toxicities, such as autoimmune-like side effects. Therefore, researchers are currently attempting to better understand the population of cancer patients who might benefit from ICIs. Immunohistochemistry staining of PD-L1 in cancers has demonstrated benefits in choosing patients with a positive response to ICIs. However, limitations exist because clinical trials have demonstrated that high PD-L1 expression and good prognosis are not correlated [73,74]. Another promising biomarker of ICI response is high microsatellite instability (MSI-H) because it can lead to the aggregation of somatic mutations [75].

Studies have also investigated the effect of germline variants on immune traits, such as NK and T cell infiltration, as well as interferon signaling. The hope is that better understanding the role that the patient’s inherited genetic background plays in cancer immunity will allow for improved population stratification for ICI therapy. A study by Shahamatdar et al. (2020) presented a pan-cancer germline analysis of immune infiltration in solid tumors and highlighted the important role that inherited variants play in influencing the immune composition of the TME and subsequent immune infiltration [76]. For example, an SNP was identified that was associated with the number of infiltrating follicular helper T cells, and over twenty candidate genes were identified that were involved in cytokine-mediated signaling. Research such as this emphasizes the crucial rule inherited variants may play in understanding predictors of ICI efficacy. In addition, a study by Sayaman et al. (2021) found that 15–20% of intratumoral variation of interferon signaling and cytotoxic cells is heritable [77]. Therefore, the research further demonstrated that germline genetics have an impact on the TME, the tumor–immune interactions, and subsequently, the usage of ICIs.

In addition to the search for biomarkers of efficacy, new immune checkpoint inhibitors are being investigated; they are often not potent enough to be used alone, however. Targets for inhibitory immune checkpoints include LAG-3 or lymphocyte activation gene-3 (CD223), which is expressed by activated T cells, B cells, natural killer cells, and dendritic cells and interacts with MHC class II [56,78]. LAG-3’s mechanism of action is not completely understood, but the interaction with MHC class II results in a decrease in T cell cytokine production and CD4 and CD8 T cell expansion. T cells in the TME, which are called tumor-infiltrating lymphocytes, overexpress LAG-3, leading to cell dysfunction, exhaustion of the immune system, and advantageous conditions for tumor proliferation [79]. Blockade of LAG-3 leads to the activation of the immune system against cancer cells and the enhancement of the effects of other ICIs [78,80]. Currently, no biomarkers exist to predict which patients may benefit [80]. However, research is looking into six main molecules; these include the monoclonal antibodies REGN3767, LAG525, FS118, BI754111, and tebotelimab, as well as IMP321, a LAG-3-Ig fusion protein.

Another immune checkpoint being investigated is T cell immunoglobulin-3 or TIM-3, a receptor expressed by NK cells, effector T cells, Tregs, macrophages, DCs, B cells, and tumor cells that promotes immune tolerance [81,82]. Research has demonstrated a strong correlation between high TIM-3 levels and worse prognosis in prostate, renal cell, colon, and cervical cancers [82,83]. The receptor’s principal ligands include phosphatidyl serine, galectin-9, and carcinoembryonic antigen-related cell adhesion molecule [79]. TIM-3 stimulation from the aforementioned ligands leads to T cell exhaustion and expansion in the TME of MDSCs. This can eventually result in tumor growth. TIM-3 blockade by monoclonal antibodies MBG453, Sym023, and TSR-022 decreases expansion of the MDSCs and increases proliferation of T cells and production of cytokines [84].

Similarly, chimeric antigen receptor (CAR) T cell therapies have emerged since 2017 as immunotherapies for hematological tumors (Figure 3A) [85]. This therapy involves the adoptive transfer of T lymphocytes that have been reprogrammed to specifically attack tumor cells by targeting CD19, CD20, and CD22, as well as the B cell maturation antigen (BCMA) [86]. CAR T cell therapies have shown potential for solid tumors as well, demonstrating rapid tumor eradication and long-lasting response, but challenges have arisen due to the complexity of the TME [86]. Due to the immunosuppressive TME, the CAR T cells struggle to penetrate the dense fibrotic stroma of solid tumors, are inadequately activated due to a lack of chemokine expression, and become exhausted. However, specific tumor-associated antigens (TAAs) for solid tumors such as mucin-1, B7H3, human epidermal growth factor receptor 2, epidermal growth factor receptor, and carcinoembryonic antigen, have been applied to CAR T cells for use in solid tumors [86]. Tumor neoantigens also have potential for utilization in CAR T cell therapies; for example, anti-EGFRvIII-CAR T cells have been utilized for high-grade glioblastoma cells [86].

In addition, tumor neoantigens, due to their specific immunogenicity, have been utilized in personalized tumor vaccines to stimulate the patient’s autoimmune system and generate an anti-tumor response (Figure 3B) [87]. Because neoantigens are personalized, these tumor vaccines are effective in inducing tumor-specific T cells without targeting normal cells. Hundreds of synthetic long peptides, RNA-, DNA-, and dendritic cell-based vaccines have reached clinical validation trials, and one successful example is a neoantigen peptide vaccine for EGFR-mutated non-small cell lung cancer [88]. As the field of cancer treatment evolves, the clinical methodologies utilized to induce tumor regression and subsequently create lasting anti-tumor immune memory will continue to advance.

## 6. Tumor Mutation Burden as a Potential Biomarker for Response to Immune Checkpoint Inhibitors

Cancer is ultimately a product of germline and accumulating somatic DNA mutations in impacted cells [89]. Between both individual tumors and discrete tumor types, the mutational frequency varies significantly [90,91]. High TMB is correlated with an increased probability of exhibiting tumor neoantigens on the HLA molecules located on tumor cells’ surfaces [2]. It is consequential that ICIs are likely to have a better response in tumors with a higher TMB; a greater mutational burden raises the chances of identification by T cells reactive to neoantigens. Subsequently, research has advocated for the utilization of high TMB as a predictive biomarker for efficacy of ICIs, which as previously stated, are only successful in a small subsection of cancer patients [92].

Studies have demonstrated that high TMB consistently selects for an improved objective response rate with ICI therapy in select cancers [10,11,91,93,94]. Hanna et al. (2018) concluded that patients with head and neck squamous cell carcinoma who were treated with ICIs had better survival with higher TMB [10]. In addition, a metanalysis by Wu et al. (2019) found that in both NSCLC and melanoma, TMB’s predictive value was significant [92]. Despite these studies, TMB is not associated with greater survival in all types of cancer. For example, Samstein et al. (2019) determined that patients with melanoma, colorectal cancer, or NSCLC had improved outcomes with ICIs if they had high TMB, but that glioma patients had worse survival with high TMB [47].

The inconsistencies with the association between specific biomarkers and ICI therapy response reinforce the need to investigate TMB, which can be used in conjunction with previously researched biomarkers including expression of PD-L1 [5]. In the Checkmate 026 trial, which compared nivolumab, an ICI, with standard of care (SOC), there was no improvement in progression-free survival (PFS) for NSCLC patients with a PD-L1 expression of less than or equal to 5% [95]. However, the trial did find a significant improvement in PFS after nivolumab treatment in comparison to SOC chemotherapy if the patient had a TMB with a minimum of two-hundred and forty-three missense mutations. These patients were in the upper TMB tertial and were defined as having high TMB. Interestingly, the patients with lower TMB had worse PFS with nivolumab treatment, which points towards lower TMB serving as a potential predictor of ICI inefficiency. In addition, in Checkmate 227, a phase III NSCLC trial, patients with high TMB had significant enhancements in PFS after nivolumab and ipilimumab treatment in combination with SOC chemotherapy in comparison to patients with positive and negative expression of PD-L1 treated with SOC chemotherapy [95].

TMB and PD-L1 expression have been shown to be independent predictive variables [94,95,96]. Carbone et al. (2017) found that in cancer patients treated with nivolumab, their TMB levels were better able to identify the beneficiaries than expression of PD-L1 [94]. On the other hand, Rizvi et al. (2018) concluded that a patient’s TMB level and PD-L1 expression value are not proportional, which points towards the possibility of utilizing PD-L1 expression and TMB in conjunction for patient screening [97]. In addition, Peters et al. (2017) concluded that consistently, greater benefit has been shown with either anti-PD-1 or anti-PD-L1 treatment if PD-L1 expression and TMB are both high [98]. These studies demonstrate the need for future research on how to best utilize TMB and PD-L1 expression as clinical biomarkers before broad use in varied tumor types.

## 7. Limitations of TMB as a Clinical Biomarker

As previously stated, additional research is needed to understand how to utilize TMB clinically due to the heterogeneity in outcomes from research analyzing the efficacy of this biomarker. This variance may result from TMB only having predictive value in specific cancer types [92]. On the other hand, different cutoff values for the TMB have been adopted, and as a result, the optimal TMB threshold is unknown for a wide range of cancer types [51,99]. Although studies generally utilize similar cutoff values for high TMB, ranging from 5 to 10 mutations/Mb, the amount of research done is insufficient to establish distinguishing criteria between low, medium, and high TMB. Due to the spectrum that exists between patients with high versus low TMB in how much they benefit clinically from ICIs, researchers have suggested for greater analysis to be performed on patients with a medium TMB because clinical guidance is lacking on how to treat them [92]. Currently, studies are investigating TMB in different cancers, and in the future, these results will allow researchers to better evaluate respective cutoffs for various types of cancers.

In addition to the varying cutoff values for the TMB, research on this biomarker has been complicated because targeted next generation sequencing (NGS) has been widely utilized due to the price and complexity of whole exome sequencing (WES) [3,100]. Two non-genome sequencing methods have been approved thus far by the FDA, MSK-IMPACT^®^ and FoundationOne CDx assay, but there are large disparities between the different gene panels utilized in research [92]. Therefore, when interpreting studies assessing the TMB as a biomarker, it is vital to consider the differences in methodologies.

In this review, it is also important to acknowledge that circulating tumor DNA (ctDNA) sequencing, a liquid biopsy method, is promising as an approach to tailoring therapy for cancer patients (Figure 3C) [100]. ctDNA profiling is noninvasive, which allows for avoidance of complications related to biopsy procedures [100]. In addition, although a tissue biopsy allows one to learn about the genomic landscape of a specific tumor site, ctDNA may provide more information about tumor heterogeneity, especially if the tumor is metastatic [100]. When a large panel is utilized, this new technology has clinical potential to routinely monitor or detect cancer, identify biomarkers of ICI efficacy, including TMB, and subsequently match cancer patients with targeted therapies [101].

## 8. Current Approaches to Overcoming Immunoresistance to ICI Treatment

Although therapeutic response to ICIs has been characterized by unparalleled durability in a minority of patients, in this review, it is critical to emphasize that the majority of treated patients are resistant to ICIs [102]. Patients can either have primary resistance to ICIs and not respond to treatment at all or can have acquired resistance and undergo a period of initial response to ICIs followed by progression of malignancy (Figure 4A). Rates of primary and acquired resistance differ across diagnoses. Due to the high rates of acquired resistance to ICIs, some researchers have looked into persistent tumor mutation burden (pTMB), instead of TMB, as a biomarker for ICI efficacy [103]. pTMB has been defined as the mutations in single-copy regions and those present in multiple copies per cell; therefore, it is a measure of “tumor foreignness” within the TMB that cannot be changed by neoantigen loss during tumor evolution [103]. Studies have found that these persistent mutations are retained during tumor evolution despite the selective pressure of immunotherapies. Therefore, it has been hypothesized that these mutations may convey sustained neoantigen-driven immune responses. In addition, research has found that tumors with a high pTMB have a more inflamed TME [103].

In terms of their TME and response to cancer immunotherapies, tumors have been separated into two main categories: hot and cold tumors (Figure 4B) [104]. Hot tumors have an immunosupportive TME and are responsive to immunotherapy, while cold tumors have an immunosuppressive TME and show a poor response to immunotherapy. An evolving area of research has focused on targeting different aspects of the TME in order to transition cold tumors to hot. For example, studies have targeted vascular normalization and proposed that endoglin (CD105) may be a useful therapeutic target due to its involvement in angiogenesis, inflammation, and CAF accumulation [104].

On the other hand, research has also focused on cancer cell intrinsic mechanisms of resistance to ICIs including loss of antigen presentation and impaired response or prolonged exposure to interferon-gamma [105]. Interestingly, a large percentage of patients are immunoresistant to ICIs due to insufficient antigen presentation to activate T cells [106]; subsequently, research has focused on the relationship between impaired MHC-I expression and immunoresistance to ICI treatment. Although interferon-gamma within tumors can increase MHC-I gene expression through the JAK-STAT pathway, therefore increasing CD8+ T cell and anti-tumor immune response, interferon-gamma also has the ability to diminish the immune response by increasing expression of CD274, which encodes for PD-L1 [105]. However, recent research analyzing methods of upregulating MHC I expression without PD-L1 has demonstrated that knocking out the TRAF3 gene, a negative regulator of MHC I, through NF-kB, increased sensitivity to T cell cytotoxicity in tumors [105]. Without TRAF3, there was an increase in antigen-presenting gene expression, as well as an overall increase in response and survival rates following ICI therapy. Furthermore, birinapant, a peptidomimetic of second mitochondrial-derived activator of caspases (SMAC), was found to also increase MHC I expression and improve response to ICIs [105].

The role of MHC class II molecules in the tumor immune response has also been an evolving field of research. Research has shown that an increased expression of MHC II within tumor cells is correlated with an increase in immunotherapy response [107]. In addition, studies have determined that MHC II is expressed at varied levels in different diagnoses and that melanoma is on the higher end of the spectrum. Following the utilization of a syngeneic transplantation model, it was found that melanoma cells with higher MHC II expression had a greater response to anti-PD-1 immunotherapy and that the Hippo signaling pathway has a vital role in controlling MHC II expression [107]. Future research could look further into modulating the pathway and whether there could be an improved response for melanoma patients to anti-PD-1 immunotherapy.

## 9. Conclusions

Currently, there is not a standard biomarker utilized clinically to predict tumor response to immune checkpoint inhibitor therapy. However, research has recently pointed towards the tumor mutation burden as a potential biomarker of ICI efficacy due to the increased production of neoantigens that accompanies a greater number of somatic mutations. TMB levels differ across patients according to the type of cancer, the tumor subtype, the patient’s age, environmental factors, and germline inactivation of specific genes. In addition, microsatellite instability, MMR deficiencies, and base excision and homologous recombination repair mechanisms can contribute to a higher somatic TMB [43,44].

Thus far, studies have demonstrated that patients with select cancers, such as melanoma and NSCLC, who were treated with ICIs, had better survival if they had a high TMB [12]. However, high TMB has not been associated with greater survival in patients with other diagnoses, including gliomas and breast and prostate cancers. In addition to this inconsistency in predictive value between diagnoses, the cutoffs utilized in each study about TMB efficacy as a biomarker vary as well. Therefore, greater research is needed to analyze the variations between cancer types and to establish cutoffs for each one [92]. Moreover, greater examination is needed for patients who fall within the “medium TMB” category as there is even greater uncertainty in the clinical field when attempting to plan patient treatment and management.

In the future, research can attempt to create a more defined and standardized method for assessing the TMB across cancer types. In terms of methodology, it is also important to acknowledge that a majority of research thus far has focused on PD-L1 therapy; this points towards the need to, in the future, also examine TMB as a biomarker of efficacy of response to different classes of ICIs, such as combined anti-PD-L1 and anti-CTLA-4 therapy [92]. TMB has the potential to be used concurrently alongside current biomarkers for ICI efficacy including PD-L1 expression, but further investigation is needed before it can be fully implemented in clinical settings.

## Figures and Tables

**Figure 1 ijms-24-06710-f001:**
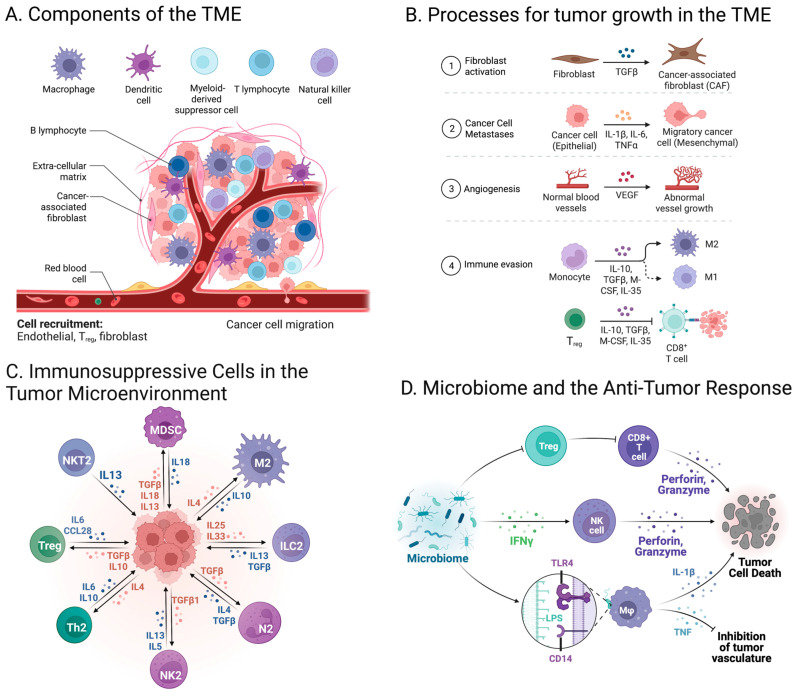
Tumor microenvironment. (**A**) The TME develops around tumor cells and is composed of the extracellular matrix, immune cells, mesenchymal stromal cells, blood vessels, and cancer-associated fibroblasts. (**B**) The tumor is able to grow and metastasize due to fibroblast activation, cytokine signaling, angiogenesis, and immune evasion. (**C**) Important immune cells in the TME include T cells, B cells, natural killer cells, and macrophages. (**D**) Research has shown that the gut microbiome has an integral role in the anti-tumor immune response and subsequently impacts a patient’s response to ICIs (Created with BioRender.com; (accessed on 15 January 2023)).

**Figure 2 ijms-24-06710-f002:**
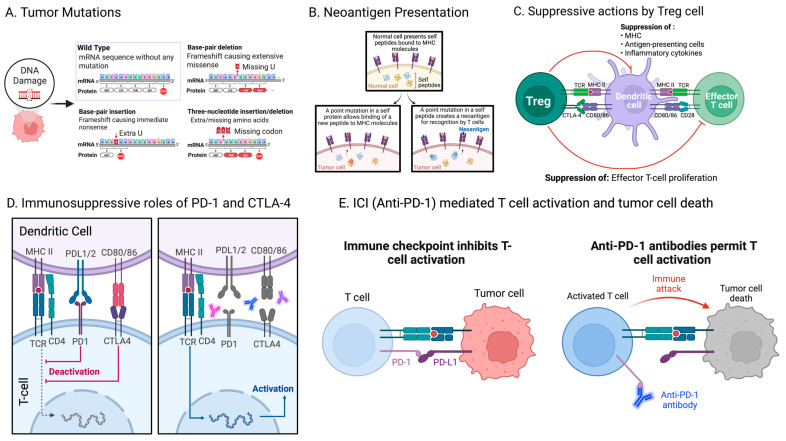
Neoantigens and current immune checkpoint inhibitors. (**A**) Tumor mutation burden (TMB) is the total number of somatic mutations found in the cancer cells’ DNA. (**B**) On the surface of tumor cells, neoantigens are presented and subsequently recognized by T cells. (**C**) Treg cells can reduce the cytotoxic T cell response against transformed cells. (**D**) Types of cancer treatment include surgery, chemotherapy, radiation, and immunotherapy. ICIs are a type of immunotherapy that promotes T cell activation. CTLA-4 and PD-1 signaling can be blocked by current ICIs. (**E**) PD-1 is an inhibitory checkpoint with an immunosuppressive role. Anti-PD-1 antibodies allow for T cell activation (Created with BioRender.com; (accessed on 15 January 2023)).

**Figure 3 ijms-24-06710-f003:**
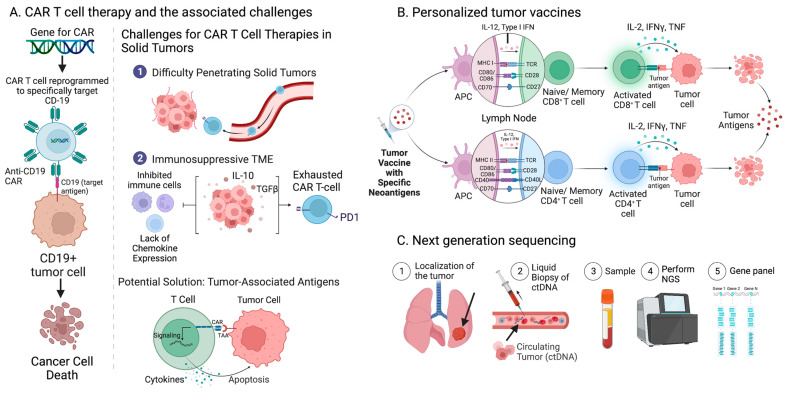
New trends in the field. (**A**) CAR T cell therapies involve the adoptive transfer of T lymphocytes reprogrammed to attack tumor cells by targeting antigens such as CD-19. Challenges have arisen for CAR T cell therapies in solid tumors due to the difficulty penetrating the dense stroma of the tumors and the immunosuppressive TME that decreases chemokine expression. A potential solution is the utilization of specific TAAs. (**B**) Personalized tumor vaccines utilize tumor neoantigens to generate an anti-tumor response and apoptosis of tumor cells. (**C**) Liquid biopsies and molecular characterization of ctDNA allow for targeted NGS with gene panels such as MSK-IMPACT^®^ (Created with BioRender.com; (accessed on 15 January 2023)).

**Figure 4 ijms-24-06710-f004:**
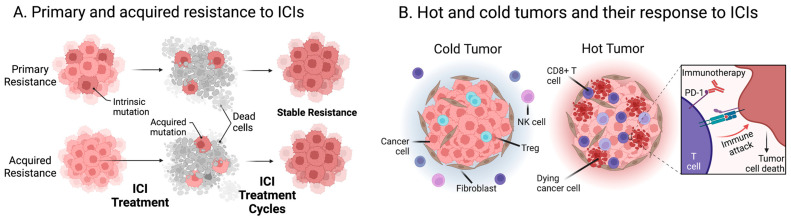
Immunoresistance to ICIs. (**A**) Patients can have primary resistance to ICIs and not respond to treatment at all or can have acquired resistance and undergo a period of initial response to ICIs followed by progression of malignancy and stable resistance. (**B**) Cold tumors have an immunosuppressive TME and show a poor response to immunotherapy, while hot tumors have an immunosupportive TME and are responsive to immunotherapy (Created with BioRender.com; (accessed on 15 January 2023)).

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
