# Peer review of "The Efficacy of Tumor Mutation Burden as a Biomarker of Response to Immune Checkpoint Inhibitors"

_ijms, 2023, doi:10.3390/ijms24076710_

Round 1

Reviewer 1 Report

Moeckel et al. in this Review focused on cancer research therapeutics, including immune checkpoint inhibitors (ICIs) and tumor mutation burden (TMB).

The Review is well organized and written and is supported by superb Figures.

Minor comments are found below:

- Abstract: In this section is missing the aim of the Review. Please, briefly specify it.

- Pag. 1, line 44: MSK-IMPACT. Please, open it for those who are not familiar with it.

- Pag. 2, lines 47-48: PD-L1. Please, open it.

- Pag. 3, line 116: If possible, I would suggest the authors to provide with more information about the possible categorization of macrophages (M1 vs M2).

- Pag. 3, line 131: This is an emerging field of research and to date there are many studies showing the importance of microbiota as drivers of inflammation. Please up-date the information in the text/references.

- Pag. 3, line 134: The same is for the mycobiome. Please, adjust it.

- Pag. 7, line 276: TME vs pag. 2, line 47. Using the same abbreviations confuses the reader. Please, modify it.

- Pag. 8, line 328: I would include also germline mutations.

- Pag. 9, lines 392-395: Please modify this paragraph because does not sound correct.

- I would suggest to increase the font size of each Figure.

Author Response

  1. Abstract: In this section is missing the aim of the Review. Please, briefly specify it.

 Author response: We thank the reviewer for the comment. We have now revised the abstract to clearly state the aim of the Review, as seen below:

… In this review, we aim to explore current research on the efficacy of the TMB as a biomarker, discuss current approaches to overcoming immunoresistance to ICIs, and highlight new trends in the field such as liquid biopsies, next generation sequencing, chimeric antigen receptor T-cell therapy, and personalized tumor vaccines.”

  1. Pag. 1, line 44: MSK-IMPACT. Please, open it for those who are not familiar with it.

 Author response: We thank the reviewer for the comment. We have included information to introduce the readers to MSK-IMPACT and have included the changes below:

TMB is clinically defined as the total sum of base substitutions in targeted genes’ coding regions; for instance, targeted tumor-sequencing panels such as the MSK-IMPACT® (Memorial Sloan Kettering-Integrated Mutation Profiling of Actionable Cancer Targets) can be utilized [4]. MSK-IMPACT® is the first laboratory-developed tumor-profiling test to receive authorization from the U.S. Food and Drug Administration. It can detect >468 gene mutations and other critical genetic changes, such as microsatellite instability, in common and rare cancers, to help the identification of those patients who may benefit from immunotherapy.”

  3. Pag. 2, lines 47-48: PD-L1. Please, open it.

Author response: We thank the reviewer for the comment. We have included information to introduce the readers to PD-L1 and have included the changes below:

Before TMB was investigated as a biomarker for ICI efficacy, research focused on programmed-death ligand-1 (PDL-1) expression in the tumor microenvironment (TME); the binding of programmed death cell receptor-1 (PD-1) to PD-L1 limits activation of T cells and subsequently decreases the immune response to cancer cells [6].”

 4. Pag. 3, line 116: If possible, I would suggest the authors to provide with more information about the possible categorization of macrophages (M1 vs M2).

Author response: We thank the reviewer for this comment. We have now included more information about the possible categorization of macrophages (M1 vs M2) and have included the updated paragraph below.

Myeloid-derived suppressor cells (MDSCs) within the TME also aid in tumor development and immune tolerance. MDSCs typically relocate to the peripheral lymphoid organs and differentiate into macrophages and dendritic cells [22]. However, in the TME, MDSCs differentiate into tumor-associated macrophages (TAMs). TAMs are categorized as M1 or M2; M1 macrophages exhibit anti-tumor responses, while M2 macrophages promote tumor growth through several mechanisms including neovascularization, angiogenesis, and modification of stromal cells for greater tumor support [23,24]. M0 macrophages are undifferentiated macrophages, but are prone to differentiate into M2 macrophages because they are recruited to the site via M2-associated cytokines [23].”

  1. Pag. 3, line 131: This is an emerging field of research and to date there are many studies showing the importance of microbiota as drivers of inflammation. Please up-date the information in the text/references.

Author response: We thank the reviewer for this comment. We have now updated this paragraph and the references to better reflect the multitude of new studies showing the importance of the microbiome and mycobiome as drivers of inflammation. The updated paragraph section is as seen below:

Studies have demonstrated that the gut microbiome has an integral role in the anti-tumor immune response and subsequently impacts a patient’s response to ICIs [27-30]. The gut microbiome’s impact on the TME includes decreasing tumor quantity, decreasing inflammation, and slowing metastatic growth [31]; studies have implicated specific gut microbial metabolites such as TMAO in driving antitumor immunity [32]. However, recent studies have also implicated specific microbiota in driving inflammation and particular forms of cancer growth [33,34]. Therefore, it is imperative for research to continue exploring the connections between the microbiome, tumorigenesis, and the response to cancer immunotherapies [28]; this research will promote future strategies to modulate the microbiome to improve outcomes to ICIs [28].”

6. Pag. 3, line 134: The same is for the mycobiome. Please, adjust it.

Author response: We thank the reviewer for the comment. We have updated this paragraph to better reflect the multitude of new studies showing the importance of the mycobiome as driver of inflammation. We have included the updated paragraph section below:

In terms of the mycobiome, a study on pancreatic ductal adenocarcinoma (PDA) has shed light recently on the impact of disbalance in the mycobiome [35]. When compared with healthy pancreatic tissue, a sample of PDA contained a 3,000-fold amount of fungi, especially Malassezia spp [35]. However, when ablation of the mycobiome was performed, there was evident protection against tumor proliferation in slowly progressing and invasive forms of PDA [35]. Lastly, the study showed that propagation with a Malassezia species specifically accelerated oncogenesis [35]. In addition, a study analyzing the presence of Candida, a fungus, in gastrointestinal cancers found that several Candida species were increased in tumor samples and tumor-associated Candida DNA was predictive of reduced patient survival [36]. The fungus was implicated in an increased pro-inflammatory immune response and linked with decreased regulation of genes involved in cellular focal adhesion and metastasis [36].

Given the evolving research on the mycobiome’s role in tumor development, it has also been recently explored as a target in cancer therapy. Administration of specific β-glucans, one of the polysaccharides in the fungal cell wall, has been shown to regulate the TME, resulting in a decrease in tumor growth and metastasis [37]. Other research has suggested that β-glucan may modulate the immune response, subsequently increasing the response to ICIs [37]. Studies such as these push forward knowledge about the TME and therapeutic treatment options for cancer patients [37].

  1. Pag. 7, line 276: TME vs pag. 2, line 47. Using the same abbreviations confuses the reader. Please, modify it.

Author response: We thank the reviewer for this comment. We have now changed this on page 7, line 276, as follows: “T cells in the TME, which are called tumor-infiltrating lymphocytes, overexpress LAG-3, leading to cell dysfunction, exhaustion of the immune system, and advantageous conditions for tumor proliferation [80].”

8. Pag. 8, line 328: I would include also germline mutations.

Author response: We thank the reviewer for the comment. We have revised the identified sentence and included the change below:

Cancer is ultimately a product of germline and accumulating somatic DNA mutations in impacted cells [90].”

9. Pag. 9, lines 392-395: Please modify this paragraph because does not sound correct.

Author response: We thank the reviewer for this comment. We have now carefully checked over the information in the paragraph against the cited sources for accuracy, and we have included the revised paragraph below:

In this review, it is also important to acknowledge that circulating tumor DNA (ctDNA) sequencing, a liquid biopsy method, is promising as an approach to tailoring therapy for cancer patients (Figure 3C) [101]. ctDNA profiling is noninvasive, which allows for avoidance of complications related to biopsy procedures [101]. In addition, although a tissue biopsy allows one to learn about the genomic landscape of a specific tumor site, ctDNA may provide more information about tumor heterogeneity, especially if the tumor is metastatic [101]. When a large panel is utilized, this new technology has clinical potential to routinely monitor or detect cancer, identify biomarkers of ICI efficacy, including TMB, and subsequently match cancer patients with targeted therapies [102].

  1. I would suggest to increase the font size of each Figure.

Author response: We thank the reviewer for their positive assessment regarding the graphical illustrations. We have now increased the size of the text in the graphical illustrations to make them easier to read

Reviewer 2 Report

The review “The effect of tumor mutation burden in the immune response of cancer patients” by Moeckel et. al is although basic, yet a well written article that tries to focus on, and introduce TMB as an important factor to consider as a biomarker for ICI therapies. The manuscript is well done, especially the quality of figures, and I just have a few small recommendations that need to be addressed. Once the authors have rectified my concerns, I am happy to accept this article for publication.

Major comment –

1.     The title of the article says “effect” of TMB in immune response of cancer patients. However, in my opinion, the article is more about introducing TMB as a potential biomarker to access immune response in cancer patients subjected to ICI treatments (wherever applicable). I recommend that the authors modify the title to introduce TMB as a biomarker in ICI therapy rather than say “effect of TMB”.

2.     The Abstract needs to have a clear message. While the first half is great, the second half of the abstract falls apart from the sentence “As a result of these findings……”. It is not clear what the authors want to say and there are typographical errors as well. Please revise the second half to make the message of the review clearer.

3.     The text in figure 1A, 1B, 1C, 2A, 2B, 2C and 3B are too small to read. These would benefit from an increase in size. Please expand these.

Minor comments –

1.     Line 197 page 5, mutations are not transcribed or translated, the genes containing those mutations are. Please rectify your statement.

2.     Line 392, Page 9. Please do not start the paragraph with “However……”. Modify the opening sentence.

Author Response

  1. The title of the article says “effect” of TMB in immune response of cancer patients. However, in my opinion, the article is more about introducing TMB as a potential biomarker to access immune response in cancer patients subjected to ICI treatments (wherever applicable). I recommend that the authors modify the title to introduce TMB as a biomarker in ICI therapy rather than say “effect of TMB”.

Author response: We thank the reviewer for the comment. We have now modified the title of our review, as suggested: “The Efficacy of Tumor Mutation Burden as a Biomarker of Response to Immune Checkpoint Inhibitors”

  1. The Abstract needs to have a clear message. While the first half is great, the second half of the abstract falls apart from the sentence “As a result of these findings……”. It is not clear what the authors want to say and there are typographical errors as well. Please revise the second half to make the message of the review clearer.

Author response: We thank the reviewer for the comment. We have now carefully revised the second half of the abstract, as requested.

Therefore, future research is needed to analyze the variations between cancer types and establish TMB cutoffs in order to create a more standardized methodology for using the TMB clinically. In this review, we will explore current research on the efficacy of the TMB as a biomarker, discuss current approaches to overcoming immunoresistance to ICIs, and highlight new trends in the field such as liquid biopsies, next generation sequencing, chimeric antigen receptor T-cell therapy, and personalized tumor vaccines.”

  1. The text in figure 1A, 1B, 1C, 2A, 2B, 2C and 3B are too small to read. These would benefit from an increase in size. Please expand these.

Author response: We thank the reviewer for this suggestion. We have now increased the size of the text in the graphical illustrations to make them easier to read. 

  1. Line 197 page 5, mutations are not transcribed or translated, the genes containing those mutations are. Please rectify your statement.

Author response: We thank the reviewer for spotting this error. We have now revised the sentence as suggested.

After transcription and translation of the genes containing the mutations, the neoantigen-containing peptides undergo processing and are subsequently presented on major histocompatibility complex (MHC) molecules on the cell surface.”

  1. Line 392, Page 9. Please do not start the paragraph with “However……”. Modify the opening sentence.

Author response: We thank the reviewer for the comment. We have now modified the opening sentence as prompted:

In this review, it is also important to acknowledge that circulating tumor DNA (ctDNA) sequencing, a liquid biopsy method, is promising as an approach to tailoring therapy for cancer patients (Figure 3C) [101].”

Reviewer 3 Report

In this manuscript," The Effect of Tumor Mutation Burden in the Immune Response of Cancer Patients," the authors described the tumor microenvironment, the role of mutations variability tumors across different tumors, and the association of mutations and immune system response. The authors have also discussed the Tumor Mutation Burden as a potential marker for response to immune checkpoint inhibitors and its limitation as a clinical biomarker. The review article is well-written and covers all essential information regarding Tumor Mutation Burden. However, the abstract does not highlight the aim of the review clearly. The abstract should clearly state this review paper's objective from the abstract to the readers.

Minor suggestions:

  1. Line number 33; please provide support to this statement.
  2. Line number 103; please give support to this statement.
  3. Provide supporting details for lines 140 and 141.
  4. Line 163; provide support.
  5. Restructure line 246.
  6. Please provide strategies to reduce tumor-burden-associated impacts on immunity.

Author Response

  1. The abstract does not highlight the aim of the review clearly. The abstract should clearly state this review paper's objective from the abstract to the readers.

Author response: We thank the reviewer for the comment. We have now carefully revised the abstract, as suggested.

Cancer is one of the leading causes of death in the world; therefore, extensive research has been dedicated to exploring potential therapeutics, including immune checkpoint inhibitors (ICIs). Initially, programmed-death ligand-1 was the biomarker utilized to predict the efficacy of ICIs. However, its heterogeneous expression in the tumor microenvironment, which is critical to cancer progression, promoted the exploration of the tumor mutation burden (TMB). Research in various cancers, such as melanoma and lung cancer, has shown an association between high TMB and response to ICIs, increasing its predictive value. However, the TMB has failed to predict ICI response in numerous other cancers. Therefore, future research is needed to analyze the variations between cancer types and establish TMB cutoffs in order to create a more standardized methodology for using the TMB clinically. In this review, we will explore current research on the efficacy of the TMB as a biomarker, discuss current approaches to overcoming immunoresistance to ICIs, and highlight new trends in the field such as liquid biopsies, next generation sequencing, chimeric antigen receptor T-cell therapy, and personalized tumor vaccines.”

  1. Line number 33; please provide support to this statement.

Author response: We thank the reviewer for the comment. We now provide support to this statement (ref#[2]), as suggested. The order of our references has therefore changed, as seen in track changes in the manuscript.

Currently, cancer is one of the top three causes of death in the world [1], but immune checkpoint inhibitors (ICIs) have demonstrated potential in treating a multitude of these malignancies, including melanoma and non-small cell lung cancer (NSCLC) [2].

  1. Line number 103; please give support to this statement.

Author response: We thank the reviewer for the comment. We now provide support to this statement (ref#[20]), as suggested.

“According to this model, tumor cells secrete hydrogen peroxide, which induces oxidative stress in the stromal cells; subsequently, CAFs undergo aerobic glycolysis in order to supply nearby cancer cells with high-energy products such as lactic acid, fatty acids, ketone bodies, and pyruvate [20].”

  1. Provide supporting details for lines 140 and 141.

Author response: We thank the reviewer for the comment. We have now revised the sentence providing supporting details, as seen below.

When compared with healthy pancreatic tissue, a sample of PDA contained a 3,000-fold amount of fungi, especially Malassezia spp [35]. However, when ablation of the mycobiome was performed, there was evident protection against tumor proliferation in slowly progressing and invasive forms of PDA [35]. Lastly, the study showed that propagation with a Malassezia species specifically accelerated oncogenesis [35]... Studies such as these push forward knowledge about the TME and therapeutic treatment options for cancer patients [37].

  1. Line 163; provide support.

Author response: We thank the reviewer for the comment. We have now revised the sentence as requested.

Later in a tumor’s evolution, mutations can occur only in select cells; these subclonal mutations have been attributed to the APOBEC cytidine deaminase family [42].

  1. Restructure line 246.

Author response: We thank the reviewer for the comment. We have now restructured this sentence, as suggested.

However, with combination strategies, there is concern for toxicities, such as auto-immune-like side effects.

  1. Please provide strategies to reduce tumor-burden-associated impacts on immunity.

Author response: We thank the reviewer for this significant comment. In order to address it, we briefly discuss the different strategies to reduce tumor-burden associated impacts on immunity, citing the paper by Kim et al. (2020) (Tumor Burden and Immunotherapy: Impact on Immune Infiltration and Therapeutic Outcomes. [PMID: 33597954]). Kim et al. (2020) briefly cover surgery, chemotherapy, and radiation therapy and how they directly reduce tumor size and modulate the TME. Here, we have added a citation to this paper and briefly elaborated on these three strategies, as prompted by the reviewer.

In the recent past, the three crucial components of cancer treatment were surgery, chemotherapy, and radiation; all three aim to reduce tumor-burden associated impacts on immunity by reducing the tumor size and adjusting the TME to hopefully alleviate immune suppression [69].

Round 2

Reviewer 3 Report

In the manuscript "The efficacy of tumor Mutation burden as a biomarker of response to immune checkpoint inhibitors," the authors have addressed all my comments. The manuscript explored potential therapeutics, including immune checkpoint inhibitors. Also, the review has discussed current approaches to overcome immunoresistance to ICI's and highlights a new trend in the field.